# P53: A Guardian of Immunity Becomes Its Saboteur through Mutation

**DOI:** 10.3390/ijms21103452

**Published:** 2020-05-13

**Authors:** Arjelle Decasa Agupitan, Paul Neeson, Scott Williams, Jason Howitt, Sue Haupt, Ygal Haupt

**Affiliations:** 1Tumour Suppression Laboratory, Peter MacCallum Cancer Centre, 305 Grattan St, Melbourne 3000, Victoria, Australia; arjelle.agupitan@petermac.org (A.D.A.); Sue.Haupt@petermac.org (S.H.); 2Sir Peter MacCallum Department of Oncology, The University of Melbourne, Parkville 3010, Victoria, Australia; Paul.Neeson@petermac.org; 3Cancer Immunology Research, Peter MacCallum Cancer Centre, Melbourne 3000, Victoria, Australia; 4Division of Radiation Oncology and Cancer Imaging, Peter MacCallum Cancer Centre, Melbourne 3000, Victoria, Australia; Scott.Williams@petermac.org; 5School of Health Sciences, Swinburne University, Melbourne 3122, Victoria, Australia; jason.howitt@florey.edu.au; 6Florey Institute of Neuroscience and Mental Health, University of Melbourne, Parkville 3010, Victoria, Australia; 7Department of Clinical Pathology, University of Melbourne, Parkville 3010, Victoria, Australia; 8Department of Biochemistry and Molecular Biology, Monash University, Melbourne 3800, Victoria, Australia

**Keywords:** cancer immunity, mutant p53, cancer immunotherapy, inflammation, gain of function

## Abstract

Awareness of the importance of immunity in controlling cancer development triggered research into the impact of its key oncogenic drivers on the immune response, as well as their value as targets for immunotherapy. At the heart of tumour suppression is p53, which was discovered in the context of viral infection and now emerges as a significant player in normal and cancer immunity. Wild-type p53 (wt p53) plays fundamental roles in cancer immunity and inflammation. Mutations in p53 not only cripple wt p53 immune functions but also sinisterly subvert the immune function through its neomorphic gain-of-functions (GOFs). The prevalence of mutant p53 across different types of human cancers, which are associated with inflammatory and immune dysfunction, further implicates mutant p53 in modulating cancer immunity, thereby promoting tumorigenesis, metastasis and invasion. In this review, we discuss several mutant p53 immune GOFs in the context of the established roles of wt p53 in regulating and responding to tumour-associated inflammation, and regulating innate and adaptive immunity. We discuss the capacity of mutant p53 to alter the tumour milieu to support immune dysfunction, modulate toll-like receptor (TLR) signalling pathways to disrupt innate immunity and subvert cell-mediated immunity in favour of immune privilege and survival. Furthermore, we expose the potential and challenges associated with mutant p53 as a cancer immunotherapy target and underscore existing therapies that may benefit from inquiry into cancer p53 status.

## 1. An Immunological Precedent for p53 Function

An interest in the involvement of p53 in cancer immunology is emerging. Beyond its key role in maintaining genomic integrity [1,2], extrinsic (non-cell) autonomous p53 functions affect the surrounding tumour microenvironment (TME) through the induction of senescence, inflammation and immunomodulatory effects [3,4,5]. While overshadowed by the role of p53 in tumour suppression, local immune regulation has long been linked to p53 dysfunction, with viral infections such as Simian virus 40 (SV40), where Large T antigen complexes with and inactivates p53, or Human Papilloma Virus (HPV), where the viral E6 proteins mediates p53 proteasomal destruction [6,7,8], showing reduced immune responses. Furthermore, the impact of mutant p53 on the immune response has been inferred from studies focusing on wild-type p53 (wt p53) functions, rather than being studied directly.

Recent studies have identified several p53 target genes and regulators that play fundamental functions in immune signalling pathways that coordinate the response to cytokine production and inflammatory response, and in innate and cell-mediated immunity. In this review, we discuss the recent finding on the role of mutant p53 in cancer-related inflammation and immunity. Moreover, we highlight the relevance of mutant p53 not only in disabling the inherent wt p53-mediated tumour immunosurveillance, but also in enhancing tumour-associated immune dysfunction as a means to promote tumorigenesis, metastasis, and invasion.

### 1.1. p53 Influences the Immune and Inflammatory Tumour Microenvironment

A fine homeostatic balance exists between the role of p53 as a tumour suppressor and its impact on immune regulation. This is exemplified by the intrinsic role of wt p53 in facilitating immunity, countered by its contribution to chronic inflammation in cancer development [9,10]. In this section, we discuss how mutant p53 impinges on components of the TME to shape a pro-tumorigenic immune landscape.

Molecular and cellular components of emerging tumours may be hostile to their microenvironment and create a tumouricidal inflammatory niche. With cancer progression, the TME can develop into an immune-suppressive nest (Figure 1). Specifically, stromal cells (blood, cancer-associated fibroblasts, vascular and lymphatic endothelial cells) and infiltrating immune cells (lymphocytes and myeloid cells) cooperate to determine the functional outcome of the TME. An inherent back-and-forth of signalling between the developing tumour and its environment can ultimately maintain and promote an inflammatory, yet immunosuppressive, environment for the tumour, in response to the aberrant secretion of signalling molecules [11,12]. The collective tumour microenvironment is thus one that hijacks inflammatory signals to remodel its surroundings, recruits tumour-associated immune cells, expresses pro-tumorigenic chemokines and cytokines, and promotes neo-angiogenesis [13,14]. Cancer is able to counter normal immune functions that are deleterious to its progression by disrupting common immune effector cell function through a reduction in FAS (Fas cell surface death, CD95) receptor surface expression, and increased expression of FAS-mediated apoptosis inhibitors [15], or by disrupting cytotoxic immune signalling [16]. Moreover, immunotolerant and immune-suppressive signals, such as the checkpoint molecule PD-L1, are utilized, or immunosuppressive cell infiltrate is recruited [17,18,19]. Cellular senescence is induced in stressed and damaged cells as a strategy to maintain tissue and cellular integrity, and it can impact on tumour development. Senescence can trigger several changes in chemokine and cytokine signalling, which are collectively referred to as senescence-associated secretory phenotype (SASP). While SASP serves initially to curb pre-malignant lesions to promote damage repair and cell clearance, the prolonged secretion of inflammatory factors can foster tumour-friendly TME development and angiogenesis [20]. These subversions of the normal immune response promote cancer to flourish in the host environment.

### 1.2. The Heated Interplay between p53 and Inflammation

Chronic inflammation stokes environmental and genetic stress, which fuels tumorigenesis [21]. Cancer development frequently coincides with a shift from acute inflammatory tumouricidal mediators, including IL-12 and IFN-γ, to those that are chronically immunosuppressive, such as IL-8, IL-17, IL-23, and IL-13. This modification of the cytokine landscape requires the activation of NF-κB and Signal-Transducer and -Activator of Transcription (STAT) pathways, not only in cancer cells, but also among the TME components. The resulting environment is rich in reactive oxygen species (ROS) generated from increased receptor signalling, metabolic activity, mitochondrial dysfunction and infiltrating immune cells, which ultimately prompts chronic stress and genotoxic signalling [22]. The SASP, which also contributes to and fuels such shifts in inflammatory signalling, is also regulated by p53, its isoforms, and protein family members, and co-opts several of the aforementioned inflammatory triggers [23,24]. The effects of SASP on cancer is mediated primarily through the inflammatory phenotype and the consequent changes in cellular signalling it produces, which have been extensively reviewed elsewhere [25,26]. Below, we discuss studies that suggest that mutant p53 may take centre stage and orchestrate this immunological shift to promote a tumour-promoting TME (Figure 1).

#### 1.2.1. The Inflammatory Crosstalk between NF-κB and p53

The NF-κB and wt p53 pathways are often considered antagonistic transcriptional networks. While the canonical role of wt p53 is growth restrictive in nature, that of NF-κB promotes cell survival and inflammation. The downstream activity of both pathways may reciprocally regulate the expression of the other. Specifically, constitutive activation of NF-κB is commonly observed in chronically inflamed and malignant tissues with the capacity to repress normal apoptotic and senescence-inducing p53 activity [27,28,29,30]. If wt p53 function is lost, aberrant inflammation can enhance tumour development [31,32]. Consistent with this, wt p53 can directly suppress the transcriptional activity of NF-κB [33,34].

Intriguingly, NF-κB and wt p53 activities can also converge to promote common outcomes. NF-κB can participate in wt p53-mediated apoptosis [35] and cooperate with wt p53 to promote senescence in IMR-90 cells but not human BJ fibroblasts [36]. NF-κB is also activated downstream of p53-inducible death-domain-containing protein in response to DNA damage [37]. Pertinently, NF-κB is essential for wt p53-dependent regulation of several pro-inflammatory genes in macrophages and monocytes to amplify responses to damage signals in tissues and inflammatory environments [38].

The balance of inflammatory signalling between p53 and NF-κB signalling is maintained by the regulatory, yet cooperative, push-and-pull from both transcriptional networks. On the other hand, mutant p53 introduces a kink in this regulatory axis. Loss-of-function p53 mutations disable the measured molecular responses in this reciprocal relationship, while gain-of-function (GOF) p53 mutations can augment pro-inflammatory and survival activities of NF-κB target genes [39,40,41]. A study by Cooks et al. demonstrated that mutant p53, in cooperation with tumour necrosis factor-α (TNFα), prolongs NF-κB activation and results in a chronic-inflammatory phenotype and development of colon carcinoma in mutant p53 mouse models [39]. These observations echo the correlation of accumulated missense p53 mutants and NF-κB activation in human colitis-associated cancer [39]. Di Minin et al. uncovered a novel cytoplasmic mutant p53 GOF in human breast cancer cells, and RAS-transformed mouse embryonic fibroblasts (MEFs), capable of altering the TNF-dependent activation of NF-κB and (c-Jun N-terminal kinase) JNK pathways through the inhibition of RasGAP Disabled 2 Interacting Protein (DAB2IP). This GOF demonstrated in R280K p53 mutant, and is predicted to occur in other hotspot mutants, which are nuclear-excluded, and hence can shift TNF-induced transcription toward products that promote cell migration and lymphocyte recruitment, thereby enabling an invasive phenotype that is resistant to TNF-induced apoptosis [40]. Mutant p53 may also interact directly with NF-κB, influencing enhancer binding in response to chronic TNFα-signalling in colon carcinomas. The simultaneous binding of NF-κB and of the R273H p53 mutant and other detected mutant forms of p53 in colorectal carcinomas regulates RNA polymerase II recruitment to these enhancers and facilitates mRNA synthesis and the activation of tumour-promoting genes like *MMP9* and *CCL2* [41]. Mutant p53 in synergy with NF-κB can thus shape the inflammatory TME, coercing both epithelial and non-epithelial cells to favour cancer-promoting gene expression [4,38,39].

Consequently, opposing the pro-tumorigenic arms of the NF-κB-p53 axis is an appealing target for cancer therapy [42]. Indeed, NF-κB inhibition to restore wt p53 function is a rational approach that has previously been demonstrated using derivatives of 9-aminoacridine in renal cell carcinomas [43], and small molecule curaxins in several cancer cell lines and mouse tumour xenografts [44]. In a mutant p53 context, wt p53 reactivation strategies could thus supplement current NF-κB-dependent treatments [45,46].

#### 1.2.2. The Reciprocal Relationship of STAT and p53 in Response to Inflammatory Signalling

STAT pathways transcriptionally regulate biological responses to cytokines, chemokines and growth factor signals alongside NF-κB [47]. Like NF-κB, STAT3 is often constitutively activated in malignant tumour cells and immune cells. In fact, STAT3 interacts with NF-κB in context-dependent manners to promote several cancer hallmark characteristics including: the inhibition of cell death, increased proliferation, survival, and inflammation [48]. STAT3, and, in some cases, STAT5 and STAT6, affect the TME by promoting immunosuppressive TMEs and inhibiting anti-tumour immunity [49,50].

Pertinently, STATs can channel the inflammatory TME to impinge upon p53 activity. Like NF-κB, STAT3 impedes p53 expression, limiting its canonical tumour suppressive function [51,52,53]. In contrast, alternative phosphorylated forms of STAT3 can upregulate p53 expression through promoter binding [54]. In a manner suggestive of a feed-back loop, wt p53 is able to reduce tyrosine phosphorylation, and thus prevent STAT3 DNA-binding activity, as demonstrated in breast [55] and prostate cancer cells [56]. This reciprocal negative regulation of the phosphorylated forms of STAT3 does not occur when p53 is mutated. Indeed, the capacity of phosphorylated or alternatively spliced STAT3 to promote p53 expression may be an anticipated cancer risk when p53 is mutated. Therefore, constitutive activation of STAT3 may be selectively present in cancer cells that harbour inactivating mutation or deletion of the p53 gene, which may enable cancer cells to escape inhibition by wt p53 pathway, particularly after DNA damage. This hypothesis is supported by the status of STAT3 and p53 in prostate (DU145 and Tsu), breast (MDA-MB-468 and SK-BR-3) and ovarian (MDAH 2774, SKOV-3 and Caov-3) cancer cell lines, which express constitutively active STAT3 and either express mutant p53 or are p53 null [56]. A recent study has also shown that the R248Q p53 mutant mediates hyperactive STAT3/Jak signalling, and ablation of this mutant is sufficient to inhibit growth and invasion of colorectal cancer cell lines [57]. Although unexplored, this study likely demonstrates the ability of mutant p53 to exert novel GOFs in cancer through the differential regulation of the STAT3 pathway in inflammatory microenvironments.

#### 1.2.3. ROS Fuels the Pro-Tumourigenic Activity of Mutant p53 in Inflammatory Environments

DNA damage-induced ROS stimulates several immune pathways, including the NF-κB and STAT pathways. Wt p53 and ROS dynamically engage in maintaining the balance of these pathways, with wt p53 monitoring and maintaining ROS at permissible homeostatic threshold levels. If exceeded, as frequently occur in chronic inflammation, elevated levels of stress-associated ROS trigger apoptotic machinery [58,59,60,61,62].

Cox-2 is induced by pro-inflammatory cytokine signatures and ROS accumulation and is overexpressed in several cancers, modulating cancer cell proliferation and apoptosis [63,64]. In response to ROS activation, Cox-2 is upregulated by and interacts with p53. The consequent interaction interferes with p53 transcription and hinders stress-induced apoptosis, while augmenting cell proliferation and hepatocyte-like stem cell differentiation [65,66,67]. Specific mutations in p53 are associated with Cox-2 overexpression and in response to ROS, favour pro-tumorigenic environmental stimuli [68]. Thus, contexts of chronic inflammatory ROS, Cox-2 inhibitors like NS-398 together with an activator of p53-dependent apoptosis, doxorubicin, present an attractive therapeutic combination, as demonstrated in wt p53 expressing normal human cells [66]. In the absence of inflammatory ROS stress, Cox-2 may positively regulate wt p53 levels. The accumulation of wt p53 in response to doxorubicin or etoposide is lower in *Cox-2* knockout MEFs due to the inhibition of ROS-mediated JNK activation [69]. Thus, Cox-2 inhibitors can antagonize the cytotoxicity of therapeutic agents, as exemplified by the suppression of doxorubicin-induced cytotoxicity and p53 accumulation by NS-398 in U2OS and MCF-7 cell lines. Moreover, it is specifically the Cox-2 and ROS-associated accumulation of wt p53 that is attenuated, while the accumulation of mutant p53 in HT-29 and MDA-MB-231 cells persists when NS-398 is used with doxorubicin [69].

Mutant p53 may also impinge on endogenous antioxidant systems by differentially regulating components of the NRF2 transcriptional program responsible for the expression of antioxidant response element-dependent genes [70]. The overall effect of this interaction promotes a pro-survival ROS response in a highly inflammatory tumour environment [71,72]. By and large, the integrity of the p53 pathway is important in managing oxidative stress associated with chronic inflammatory TMEs.

### 1.3. Mutant p53 Supports and Alters Components of the Tumour Milieu

In the contexts discussed thus far, mutant p53 ultimately harnesses growth factor, chemokine and cytokine production as an instrument of pro-inflammatory, yet immunosuppressive molecular signalling. Mutant p53 has been reported to induce CXCL5, CXCL8, and CXCL12, which are pro-angiogenic and pro-invasive chemokines, which are implicated in cancer and several inflammatory diseases [73,74]. Mutant p53 (R175H, R273H, R280K) also upregulates ID4, a post-transcriptional regulator of several pro-angiogenic and tumour-supporting cytokines [75]. The mutant p53-mediated suppression of anti-inflammatory signals can also promote the tumour’s pro-inflammatory effects. For example, the suppression of the anti-inflammatory cytokine sIL-1RA by mutant p53 results in chronic inflammation associated with tumour progression [76]. Overall, mutant p53 exposed to inflammatory TMEs can form an inflammatory feed-forward loop that affects not only its encompassing cancer cell population and those surrounding it, but also the tumour stroma, extracellular matrix (ECM) and associated immune cells infiltrate (Figure 2) [5,77].

#### 1.3.1. Cancer-Associated Fibroblasts Work with Altered and Mutant p53 to Promote Inflammation and Tumorigenesis

A complex role for p53 in cancer immunity and inflammation is emerging from studies of its alteration in tumour stroma. Cancer-associated fibroblasts (CAFs) are an integral part of the TME and are heavily involved in receiving and conveying signals for inflammation and leukocyte recruitment [78,79,80,81]. CAFs in contact with cancer cells can undergo the activation of their IFN-β pathway, which cooperates with wt p53 in fibroblasts to suppress tumour growth, respond to stress and prevent cancer cell migration (Figure 2A) [82,83].

Altered p53 status in CAFs consequently affects the tumour inflammatory milieu. Cancer cells are able to suppress wt p53 activity, or rewire intact p53 pathways to mutant p53-like cancer-promoting states in associated stroma [84,85,86,87]. The resulting p53 dysfunction in CAFs alters the molecular crosstalk between the tumour stroma and cancer cells through the enhanced production of CXCL12, IL-6, and SDF-1 to promote inflammation and oncogenic signals (Figure 2B) [3,88,89,90]. The loss of wt p53 activity in CAFs can further exert selective pressure to promote transformation in neighbouring epithelial cells through ablation of p53-dependent senescence programs and skewed macrophage polarisation (Figure 2B) [91]. Less phosphorylated forms of wt p53 in lung-derived CAFs have been shown to support CAF-like properties such as migration when compared to normal fibroblasts, in part due to altered mutant-like conformation despite being genetically wild-type. Wt p53 in these CAFs also supports an altered secretome, which facilitates ECM degradation, migration and invasion [92]. Cancer cells can “educate” normal fibroblasts gradually to adopt CAF-like properties [93]. In this re-education process of normal fibroblasts, the wt p53 transcriptional program is altered upon co-culture with H460 or H1299 cancer cell lines [92]. The activity of this mutant-like wt p53 population in CAFs has profound ramifications that limit the use of treatments, which depend on canonical wt p53 activities.

While relatively uncommon, p53 mutation may occur in CAFs in hereditary cancers [94,95], in which case the surrounding microenvironment is primed for tumour formation [96]. In sporadic cancers, p53 mutation in fibroblasts occurs independently of that in tumours and is potentially clonal in origin. These mutant stromal cells potentially provide a favourable microenvironment for tumour spread [94]. In a study of bladder cancer, these stromal mutations act as neoplastic seeds for urothelial carcinoma [97]. The loss of wt p53 and alterations in its expression in CAFs further promotes ROS accumulation and alters the properties of the tumour ECM [92,98,99], potentially affecting the surrounding tumour milieu.

In contrast to its wild-type counterpart, mutant p53 in cancer cells modulates and prevents the tumour-suppressive response to IFN-β, by inhibiting STAT1 phosphorylation and downstream targets of IFN-β. In turn, IFN-β secreted by CAFs can reduce mutant p53 RNA levels in tumours (Figure 2B). This regulatory network employed by CAFs constitutes a molecular standstill that both limits and promotes the tumorigenic effects of mutant p53 in cancer cells, the balance of which can be tipped by the inflammatory microenvironment [100]. The mutational status of p53 is thus paramount to directing IFN-β-related therapies, and the reactivation of wt p53 activity may constitute a synergistic opportunity for these therapies [100].

#### 1.3.2. Mutant p53 Favours Neo-Angiogenesis and Extracellular Matrix Remodelling

In the TME, the ECM is altered to favour the infiltration of specific subsets of tumour-supportive immune cells and the formation of neo-vasculature, thus participating in what can be considered figurative inflammatory terraforming (Figure 2) [101]. Previous studies have demonstrated that the ECM can exert a regulatory effect on the p53 activity of cultured cells through pro-survival signals that suppress its apoptotic functions [102,103]. The reciprocal role of p53 in regulating the ECM has been fleshed out in recent years particularly in hypoxic contexts [104,105]. Hypoxic tumour conditions go hand-in-hand with ROS accumulation and inflammation [106]. Transcription factors that are activated in such environments play a large role in shaping the TME. In hypoxic conditions, hypoxia-inducible factors (HIFs) can induce the expression of pro-angiogenic factors including vascular endothelial growth factor (VEGF) and platelet-derived growth factor (PDGF) [107]. Furthermore, HIFs are also overexpressed in supportive cells of the TME and regulate their activity [108].

Wt p53 is able to promote the degradation of HIF-1α through MDM-2-mediated ubiquitination. Consequently, in the absence of p53, HIF-1 levels and its downstream transcriptional targets increase [109]. It has recently been demonstrated that R273H and R246I p53 mutants cooperate with HIF-1 in non-small cell lung cancer cells to transcriptionally regulate ECM components, favouring aggressive invasion and poor clinical prognosis (Figure 2B) [110]. This activity of HIFs is important for the expression of chemo-attractants to recruit supportive cells and illustrates how mutant p53 regulates these aspects of the TME [111].

Wt p53 also negatively regulates extracellular matrix metalloproteinase inducer (EMMPRIN), which is known to increase the production of several matrix metalloproteinases (MMPs) responsible for ECM remodelling, angiogenesis, and mediating tumour cell–macrophage interactions [112,113,114,115,116,117]. The down-regulation of EMMPRIN by wt p53 is not transcriptionally-dependent but is sensitive to the inhibition of the endosomal pathway by chloroquine [115]. As a corollary, p53 dysfunction can lead to EMMPRIN upregulation, and thus ECM remodelling in highly invasive cancers. Indeed, there is a correlation between mutant p53 and EMMPRIN expression in intestinal and diffuse-type gastric carcinoma, which hint at a potential GOF for mutant p53 in this context [118]. Human melanoma cells expressing mutant p53 overexpress MMP2, which is ablated upon introduction of wt p53 [119]. MMP2, in turn impinges on anti-tumour immunity [120]. The tumour ECM affects the recruitment of various cellular components of the TME, which can promote tumour cell growth as discussed below.

#### 1.3.3. P53 Status Influences the Tumour Immune Cell Infiltrate

The abundance and composition of the immune cell compartment of the TME vary among different tumour types and can contribute to the rate of disease progression and prognosis [121,122]. The immune landscape of the TME is modulated by the crosstalk between macrophages, dendritic cells, myeloid-derived suppressor cells (MDSCs), T cells, mast cells and natural killer (NK) cells (Figure 2A) [18,123]. Through sustained NF-κB and STAT3 signalling, the impairment of the epithelial barrier, and transcriptional activation of CXCL17, the loss of wt p53 activity can increase macrophage, neutrophil, monocyte and CD4^+^ T cell infiltration, while limiting the infiltration of potent anti-cancer CD8^+^ T cells (Figure 2B) [124,125,126,127,128].

In addition, mutant p53 can support myeloid cell infiltration through NF-κB-dependent inflammatory cytokine signatures [39]. The R280K p53 mutant has also been demonstrated to act in accordance with TNFα to elicit a chemokine signature that modulates the immune cell infiltrate [40]. Lymphocyte metagene analysis of tumours harbouring this p53 mutant identified a higher expression of cytotoxic T cell lymphocytes, NK cells, and Th1 genes characteristic of a pro-inflammatory immune cell signature [122]. Despite promoting cancer motility and survival, high levels of pro-inflammatory immune infiltrate correlate with better disease-free survival in patients with mutant p53 basal breast cancers [40]. This illustrates how p53 status in cancer cells can affect the clinical prognosis of cancer, depending on the nature of the immune cell landscape [127,129].

## 2. Mutant p53 Disrupts Innate Tumour Immunity

Innate immunity is the first line of defence to engage immediate short-term immune operations against pathogens without establishing immunological memory. This similarly applies to the host’s initial response to the danger signals from a developing tumour and cancer-associated inflammation [130]. Key players in innate immunity include cells of the myeloid lineage that mount effector responses in addition to priming further adaptive immune responses [131,132]. Changes in the composition of cancer cell surface proteins, and, in some cases, secreted tumour antigens, enable the activation of both arms of the innate immunity: the complement system and the toll-like receptor (TLR) system [133]. Furthermore, crosstalk between the complement and TLR pathways makes innate immunity a dynamic and robust network for first-line pathogen response [134,135].

The innate immune system is an active participant in immunosurveillance against cancer cells. Abnormal cells may, however, undergo immunoediting in response to selective pressure exerted by innate immune cells like natural killer (NK) cells, which, in concert with adaptive immune cells, kill immunogenic clones selectively. These abnormal cells evolve to evade eradication, escaping immune control and eventually developing into clinical tumours and malignancy. Identifying pathways driving the evasion of the innate immune system in the tumour context offers scope for fine-tuning therapies to reprime this intrinsic cancer defence system. Several cellular components of the inflammatory TME modulate tumour innate immunity [136]. In this section, we discuss how mutant p53 influences innate immunosurveillance in cancer.

### 2.1. P53 Mediates the Genotoxic Stress Response of the Toll-Like Receptor Pathway

Several studies hint that wt p53 acts through an expansive integrated network that mediates host intrinsic tumour immunity. Moreover, several studies demonstrate that the role wt p53 plays in cancer inflammation is not exclusive to tumour contexts and instead may be a general regulatory mechanism that aids in host responses to infection. Wt p53 expression is induced in response to viral infection as part of the host immune defence, and its role in host antiviral response has been extensively reviewed [137,138,139,140,141]. P53 is inactivated by viruses, regardless of their tumourigenic potential, to prevent host cell-mediated apoptosis [142,143]. Interestingly, the activation of wt p53 in response to cancer echoes the mechanisms employed in antiviral response, and, as such, a role for wt p53 in innate tumour immunity has been explored and developed over the years [82].

Wt p53 and the Toll-like receptor (TLR) pathways are directly linked in humans [144]. Wt p53 regulates the expression of seven out of the 10 TLRs [145]. TLRs are endosomal and plasma-membrane associated receptors that are expressed not only on immune cells, but also on cells of the TME [146]. TLRs recognize pathogen- and damage-associated molecular patterns to promote a protein kinase cascade that induces the expression of inflammatory cytokines and interferons. Additionally, TLR signalling can activate and increase natural killer (NK) cell cytotoxicity, as well as extrinsic and intrinsic apoptotic pathways in tumours [147]. Several pro-angiogenic and growth-promoting factors lie downstream of TLR signalling pathways that function in normal acute immune responses to boost the activity of antigen-presenting cells (APCs) and effector T cell responses [148,149]. Depending on the subset of immune cells activated by these factors, their response can be modulated to promote cancer [146,150,151,152].

TLR activity is canonically modulated, not by the increased expression of its member receptor proteins, but rather by the upstream stimulation of relevant receptors [153]. Though the transcriptional control of TLRs has been described, the regulation of the differential expression of TLRs in cells is commonly epigenetic in nature, and largely an indirect effect of external environmental stresses [148,154,155,156]. Thus, the discovery that several TLRs are transcriptionally regulated by a central stress-response factor, p53, constitutes an important direct link between common cellular stresses and induction of the TLR innate immune response [139,144,157].

Wt p53 regulates the expression of the TLR gene family in T-lymphocytes, and to a lesser extent in macrophages, in a manner dependent on the genetic stress and the host genetic context [144,158,159]. In particular, polymorphisms in p53 response elements in the promoters of TLR genes confer differential sensitivity to genetic stress and infection [157].

The anti-tumour benefits of such TLR induction are apparent when considering the merit of APC reactivation in the tumour microenvironment, whereby activated TLR pathways increase immune detection and activity against tumour-antigen bearing cells [160]. However, TLR expression in tumour cells and surrounding cells can prove pro-tumorigenic [161,162,163,164]. TLR4 is expressed in several human cancer cell lines, such as MDA-MB-231, MCF7, A549 and H1299. Moreover, TLR4 is functional in A549 and H1299, where it activates the p38 MAPK and NF-κB signalling pathways upon exposure to LPS treatment. This activation promotes tumour immune escape and resistance to apoptosis through production of immunosuppressive cytokines like VEGF, TGF-β and IL-8 [165]. The activation of MAPK and NF-κB are common threads in TLR-4-expressing colorectal cancers, increasing proliferative potential, apoptotic resistance and metastatic potential [166]. Moreover, TLR-4 expression in breast cancer correlates with poor survival rates and invasiveness [167,168].

Additionally, the upregulation of TLR1-10, NF-κB and p53 expression is observed in oral lichenoid disease, an auto-immune disease associated with chronic inflammation and malignant potential owing to atypical lichenoid lesions in cases of greater chronic inflammatory response [169,170]. The downstream generation of ROS and activation of NF-κB can indeed amplify the innate immune response, but may also act to generate abnormal patterns of inflammation, which, when chronically activated, prove to be pro-tumorigenic, as previously discussed [171,172]. This has a profound implication on genotoxic and TLR agonist strategies for targeting tumours, whereby enhancing TLR-signalling may instead foster a highly inflammatory yet immunosuppressive TME.

Notably, the p53-TLR regulatory axis exists only in primates and humans [144]. This evolutionary distinction is of prime consideration when evaluating TLR-mediated cancer therapies, as mouse models do not recapitulate the regulatory axis that exists in humans [162].

#### Mutant p53 Hijacks TLR Signalling in Cancer

The direct transcriptional regulation of TLR proteins by wt p53 does not necessarily reflect a linear increase in the respective ligand-dependent cytokine responses. This demonstrates the highly context-dependent nature of p53 immune signalling. Downstream TLR activity is also not solely transcriptionally dependent on p53. Target genes that crosstalk between the two pathways can cooperate to mediate and control downstream immune signalling [159]. Polymorphisms on the p53 gene and its TLR targets may contribute to the variability in downstream TLR responses observed. Indeed, single nucleotide polymorphisms in wt p53 response elements on the promoter of TLR8 modulate the wt p53 regulation of downstream immune response. Some p53 mutants retain functionality in regulating TLR expression, while several others demonstrate altered TLR regulatory spectrums. Not surprisingly, transcriptionally inactive p53 mutants, which constitute the majority of p53 mutations, are unable to mediate the upregulation of TLRs in response to genetic stress [144,158,159], supporting a role for the p53-TLR axis in the cellular response to genotoxic stress.

Notably, the clinical relevance of TLR-4 in breast cancer has been shown to be p53-dependent. In the presence of mutant p53, patients with a low expression of TLR4 correlate with better survival than those with high expression levels, whereas the inverse is true in patients with wt p53 tumours. This clinical distinction is attributed to IFN-γ secretion in wt p53 breast cancer cells that mediates growth inhibition by TLR-4. Strikingly, functional TLR-4 retention correlates with a broad spectrum of cancers with functional p53 loss, including serous ovarian, head and neck, and bladder cancers. This indicates a selective advantage to retaining TLR-4 signalling in the absence of wt p53. Inversely, tumours that retain wt p53 function correlate with a higher incidence of TLR-4 alterations, as exemplified in lung cancers [173,174].

P53 mutation not only leads to the differential expression of the TLR genes, but also impose altered downstream effects. These impact the sensitivity and responsiveness of TLR3 to its known ligands, affecting the downstream type I interferon response and genotoxic-stress induced apoptosis. This modulation of TLR3 responsiveness is directly attributed to the presence of transcriptionally active or TLR3-enhancing p53 mutants like P151H and R337H, while other mutants may conversely dampen the TLR3-mediated immune response [145,175,176]. Knowledge of TLR responsiveness to specific p53 mutants prior to the application of TLR agonists may be appropriate for enhancing the efficacy of its anti-tumour therapy. In the same vein, the restoration of p53 activity proves valuable in combination with TLR agonists to rescue TLR activation and responsiveness effects [159,161].

Mutations in p53 may also have indirect effects on TLR signalling owing to GOF effects. Arf6 and its downstream effectors are major targets for induction by p53 mutations in epithelial cells, pancreatic ductal adenocarcinoma, and mevalonate pathway-driven malignancies [177,178,179]. Strikingly, Arf6 is crucial for downstream TLR signalling and its overexpression and activity has been associated with increased proliferation, invasion and metastasis [180,181,182,183,184]. Arf6 lies in the crossroad of these two pathways through its roles in ROS production that affect both p53 and TLR pathways [58,185]. Moreover, Arf6 activation is facilitated by the platelet-derived growth factor (PDGF), a product of TLR4 signalling, and its β receptor (PDGFR_β_), a direct target of mutant p53 [178,186,187]. The pro-tumorigenic effects of Arf6 in cancer have been largely attributed to its roles in receptor recycling, matrix disruption, manipulation of cellular adhesion, aberrant growth-factor signalling, and metabolic dysfunction [180]. Arf6 may further stand as a functional link between mutant p53 and the regulation of innate tumour immunity.

### 2.2. Mutant p53 Favours a Pro-Tumour Macrophage Signature

Macrophages are one of the most abundant immune cell populations in the TME [188]. When macrophage activation is driven by lipopolysaccharide and IFN-γ, they tend toward an M1-like phenotype, which is pro-inflammatory, inducing cytotoxic responses against cancers and pathogens. In contrast, IL-4 or IL-13-mediated macrophage activation tends to elicit an M2-like anti-inflammatory phenotype, associated with wound healing and tumour promotion [189,190]. The activity of both macrophage responses is integral to several steps of tumour progression. More specifically, M1-like polarized tumour-associated macrophage (TAM) responses mediate an inflammatory environment that imposes selective pressure against highly immunogenic cancer cells driving their elimination. If the tumour escapes immune detection, the TME induces an M2-like polarization of TAMs, which favours immunosuppression and pro-tumorigenic activity [190,191].

It is generally accepted that both M1-like and M2-like polarisation are associated with increased levels of p53 expression to varying degrees. High-levels of p53 activity are elicited as a brake in M1-like macrophage polarisation to inhibit detrimental prolonged activation of the inflammatory NF-κB and STAT1 pathways, resulting in diminished M1-like gene expression over time [192]. For example, in response to iron overload, increased ROS levels enable prime p53 acetylation, while driving M1-like polarisation [193]. The elicited activity of wt p53, present only in low levels in M2-like polarized cells, permits M2-like gene expression and phenotype. The inactivation or mutation of p53 in M2-like macrophages can result in increased M2-like gene expression [194].

Wt p53 not only regulates macrophage polarisation but also impacts on the resulting inflammatory gene expression of macrophages. In macrophages, the usual antagonistic relationship of NF-κB and wt p53 is reoriented, instead forming a cooperative relationship to promote inflammatory response to genotoxic stress [38]. Demonstrating this, the Nutlin-3-mediated stabilisation of wt p53 can enhance NF-κB function in macrophage populations, amplifying early pro-inflammatory macrophage signalling. This is in stark contrast to the anti-tumorigenic effects Nutlin-3 has upon the stabilisation of wt p53 in epithelial cells, further highlighting programmatic differences in the p53-NF-κB pathway in macrophage populations [34,195,196]. Much like the TLR/p53 axis, this distinct regulatory program is absent in rodents, which, aside from having evolutionary implications, could affect the evaluation of its biological consequences using mouse models [38]. Stabilised wt p53 in macrophages potentially activates pro-inflammatory gene expression, but may also allow macrophages to adopt a senescence secretory phenotype to ensure continuous turnover in the TME through the p53 pathway [197].

Mutant p53-expressing cancers can exert similar non-cell-intrinsic reprogramming of macrophages into TAM-like M2 phenotypes through the exosomal transfer of microRNA [198]. Exosomal transfer of microRNA is functionally relevant in several cancers and may thus constitute an additional mutant-specific GOF of p53 [198,199,200]. Strikingly, exosomes from R248W and R273H mutant p53-expressing colon cancer cells are enriched for miR-1246, a microRNA found to promote invasiveness and stemness [201,202,203,204]. Following treatment of M2-like macrophages with these exosomes, genes characteristic of the tumour-supportive TAM phenotype were upregulated, demonstrating a mechanism by which mutant p53 cancers can alter innate immune cell function to promote tumorigenesis [198].

## 3. Mutant p53 Alters Cell-Mediated Immunity in Cancer

Cancer immunoediting relies on three component phases: elimination, equilibrium, and escape. The activities of cells of the adaptive immune system are also central in cancer immune editing [136]. Immunotherapy directed to tumour specific antigens relies on the integrity of APCs [130]. However, these immunogenic APCs are scantly detected in tumours, and, if are present, they display impaired or dysfunctional antigen-presenting activity. While p53 mutations are less common in immune cells, p53 can influence cell-mediated immunity through specific molecular signatures brought about by the tumour or stromal cells, which in turn affect recruitment and activation of immune cells [4]. It can also regulate the expression of class one major histocompatibility complex and associated downstream immune effects [205].

P53 has recently been shown to influence the differentiation of monocytes from already abundant myeloid precursor cells in the periphery of tumours [206]. P53 also modulates NKG2D-mediated NK cell activity through pathways in senescence. The mutation or loss of p53 in these contexts impairs NK cell-mediated immunity [138,207].

Interestingly, the role of wt p53 in inducing regulatory T cells to suppress autoimmunity hints at a mechanism by which aberrant activation of these pathways can be misappropriated in cancer [208]. Nutlin-3a-induced wt p53 reactivation in a TME rich with tumour-infiltrating leukocytes, such as in EL4 tumours, is sufficient to induce antitumor immunity, in contrast to B16 with low infiltration. This is due to an increase in the activated dendritic cell population that elicits expansion of CD8^+^ cytotoxic T cells and induces an immunogenic cell death. Indeed, the efficacy of the reactivation is p53-dependent, not only in the tumour but also in the leukocyte population within the TME [46]. Consistently, mutant p53 fails to trigger and can inhibit the tumour antigenicity that is normally caused by genomic instability, or by the prolonged activation and accumulation of wt p53, as demonstrated in gastric cancer [209]. Lymphocyte invasion, particularly that of cytotoxic T cells, is also impaired when wt p53 pathways are compromised in ER-negative breast cancer and basal-like breast tumours. Both the loss of heterozygosity and p53 mutation show lower rates of T cell infiltration and correlate with poor prognosis [210]. The loss of p53 in several genetically engineered mouse breast cancer models resulted in increased inflammatory Wnt signalling in tumour-associated macrophages, prompting systemic neutrophilia and ultimately metastasis [211]. A study using a murine melanoma model has also highlighted the requirement of p53 stabilisation for the assembly of endosomal sorting complexes that mediate immunosurveillance within the metastatic niche [212]. These studies highlight a role for wt p53 in facilitating productive tumour immunity that is compromised when p53 is lost or mutated.

More recently, connections with wt p53 and immune checkpoints have been uncovered. P53 transactivates programmed death-ligand 1 (PD-L1) and its receptor programmed death-1 (PD-1) in cancer cells, and in normal T cells in response to stress [213,214]. Physiologically, the interaction between PD-L1 on tissue and PD-1 on T cells suppresses activation signals generated following the T cell receptor recognition of antigen; this immune checkpoint controls inflammation. The overexpression of PD-L1 on tumours, however, takes advantage of this immune checkpoint pathway to suppress tumour recognition and induce immune tolerance [215]. Similarly, *FOXP3* is induced by wt p53 in breast and colon cancer cell lines in response to DNA damage. Ectopic FOXP3 expression in turn converts normal T cells to T regulatory cells, which promote an immunosuppressive tumour milieu [216]. A study using h3T T cell receptor mice with human tyrosinase epitope-reactive T cells showed that p53 knockout T cells actually served to augment anti-tumour functions. While this increased robustness was associated with decreased ROS production, which increased T cell longevity, the authors imply a role for p53-dependent FOXP3 in the observed phenotype [217]. These examples illustrate cases wherein intact wt p53 pathways are subverted by cancers to develop tumour tolerance to adaptive immunity.

More recently, connections with wt p53 and immune checkpoints have been uncovered. P53 transactivates programmed death-ligand 1 (PD-L1) and its receptor programmed death-1 (PD-1) in cancer cells, and in normal T cells in response to stress [213,214]. Physiologically, the interaction between PD-L1 on tissue and PD-1 on T cells suppresses activation signals generated following the T cell receptor recognition of antigen; this immune checkpoint controls inflammation. The overexpression of PD-L1 on tumours, however, takes advantage of this immune checkpoint pathway to suppress tumour recognition and induce immune tolerance [215]. Similarly, *FOXP3* is induced by wt p53 in breast and colon cancer cell lines in response to DNA damage. Ectopic FOXP3 expression in turn converts normal T cells to T regulatory cells, which promote an immunosuppressive tumour milieu [216]. A study using h3T T cell receptor mice with human tyrosinase epitope-reactive T cells showed that p53 knockout T cells actually served to augment anti-tumour functions. While this increased robustness was associated with decreased ROS production, which increased T cell longevity, the authors imply a role for p53-dependent FOXP3 in the observed phenotype [217]. These examples illustrate cases wherein intact wt p53 pathways are subverted by cancers to develop tumour tolerance to adaptive immunity.

In some contexts, mutant p53 can mediate increased immunogenic activity. In mutant p53-expressing breast cancer, tumours displayed a higher enrichment of immunogenic activity than those expressing wt p53. Moreover, these observations have been recapitulated in vitro, where mutant p53 regulation of cell cycle, apoptosis, Wnt, JAK-STAT, NOD-like receptor and glycolysis pathways were found to promote immunogenicity of cultured breast cancer cells. Thus, in some contexts, mutant p53 could potentially be a useful biomarker for immunotherapy responsiveness and can be associated with better survival prognosis owing to unique immunogenic signatures [218].

## 4. Mutant p53 as a Tumour Antigen

Mutations in tumour suppressor gene products can lead to the formation of novel epitopes not normally presented in self-tissues, and can thus be considered tumour antigens [219]. Mutant p53 that accumulates in cancers belongs to this class of tumour antigens. Antibodies against p53 can be detected in patient sera across several cancer types and these are strongly correlated with p53 alteration and over-expression [220,221,222,223]. The presence of p53 antibodies has been used as an early marker for diagnosing pre-malignant disease and early stages of cancer; however, the prognostic and diagnostic values of mutant p53-associated expression of p53 antibodies in sera is limited and differs depending on the type of cancer [220,224]. Mutant p53 has been considered as an appealing tumour-specific antigen target for immunotherapy, but limited success has been achieved in this field due to the inefficient presentation of mutant p53 antigen on cells for recognition [225,226,227,228].

Mutant p53 can bind to human leukocyte antigen (HLA) class I molecules to enable cytotoxic targeting by CD8^+^ T cells [229,230,231,232,233], but can also bind to HLA class II molecules [234]. The latter has been demonstrated to skew the CD4^+^ T cell response to adapt a pro-tumour Th2 phenotype in head and neck cancers [235]. Despite this, there is a low correlation with accumulated levels of p53 and their recognition by targeted T cells. In fact, the mutational status of p53 has been shown to affect its detection by human T cells. Consequently, p53-bearing destabilizing mutations like R175H and Y220C favour antigen presentation to T cells, eliciting distinct immune reactivity independent of expression levels [236]. Indeed, several studies explore the potential development of mutant p53 cancer vaccines that take advantage of this immunogenicity [237,238,239].

Recently, a screen for T cell responses against the naturally processed neoantigens of several hotspot p53 mutants (R175H, Y220C, G245S, G245D, R248L, R248Q, R248W, R249S, R273C, R273H, R273L, and R282W) revealed broad immunogenic responses that varied across patients with different metastatic epithelial cancer types and different p53 mutational status [240]. Moreover, the HLA alleles involved in the presentation of different mutant p53 antigens varied from patient to patient, highlighting a context-dependent threshold for mutant p53 antigen presentation. Despite this, HLA alleles that were capable of eliciting mutant p53 immunogenic response were frequent across several racial patient demographics, indicating a broad potential benefit for p53-neoantigen therapy. A similar screen was conducted on mutant p53 neoantigens in ovarian cancers [241]. While not all hotspot p53 mutations were immunogenic, those that were (G245S and Y220C) elicited mutant-specific T cell infiltration of ovarian cancer metastasis, further emphasizing the potential for mutant p53 as a target for T cell immune and gene therapy. Interestingly, neoantigens that arose from random, somatic, non-synonymous mutations were also recognized by T cells and were unique to each patient. Elevated levels of p53 associated with its mutation are linked to the generation of anti-p53 auto-antibodies, underpinning the potential role of p53 in mediating tumour antigenicity [220,221,228,235,242]. Consequently, fragments of p53 proteins have since been used as tumour-associated antigens for the generation of therapeutic vaccines [243,244,245,246].

Despite the promise that mutant p53 holds in the field of immunotherapy, the selective pressure that T cells exert could lead to eventual therapeutic resistance. Recent studies have employed a fusion of mutant p53 transduced dendritic cells and antigen-expressing tumour cells to generate a broad-acting vaccine, obtaining results that could potentially overcome such a pitfall [247]. The development of targeted immunotherapies against mutant p53 has proven a challenging feat, but one that potentially can reap great outcome considering the multifaceted nature of mutant p53 tumorigenic activity and its prevalence in human cancers.

## 5. Wild-Type p53 in Post-Apoptotic Cell Clearance

The post-mortem fate of cells targeted by effective anti-tumour activity also constitutes an intriguing aspect of tumour immunity and inflammation. The normal turnover of cells during the resolution of injury and infection is key to preventing aberrant inflammation and triggering the immune recognition of dead cell antigens [248]. Wt p53 further extends its role in immunity to mediating macrophage clearance of dead cells by inducing the expression of Death Domain 1α (*DD1α,* also named VISTA), a gene implicated in triggering the immune checkpoint [249]. The presence of DD1α on the surface of dying cells specifically facilitates their recognition and subsequent phagocytosis by macrophages. Inflammation due to the failure to recognize and resolve dead cell clearance may explain the resulting autoimmune phenotype associated with DD1α disruption. Similar interactions can occur between macrophage DD1α and T cell DD1α, rendering them susceptible to immunosuppression and preventing the further recognition of tumour antigens [213,250]. The role of p53 in modulating DD1α expression highlights both its tumour suppressive function and its role in modulating tumour surveillance. While this pathway is wholly reliant on intact wt p53 function, the prospect of this pathway failing, or being hyperactive in mutant p53 contexts may reinforce targeting DD1α as an effective cancer therapy.

## 6. Conclusions

The body of knowledge implicating mutant p53 in disruption of immunity that normally limits cancer control is ever increasing. A deeper understanding of the intrinsic roles of wt p53 in the TME illuminates the subversive activities of its mutant counterparts. It is now established that, aside from functional loss of p53, mutant p53 GOF contributes extensively to promoting immune and inflammatory hallmarks of cancer by (1) responding to and fuelling inflammatory signalling in the TME, (2) shaping the tumour milieu to facilitate cancer progression, (3) disrupting innate tumour immunity, (4) modulating the activity of infiltrating immune cells, and (5) potentially impinging on post-apoptotic cell clearance (Figure 3).

These proposed roles for mutant p53 predict its relevance as a target for immunotherapy as a means to limit tumour growth and metastasis. Pertinently through, the activity of mutant p53 in tumour immunity is largely context dependent. The nature of its genetic mutations, the cellular compartment in which the mutations function, as well as the molecular and cellular conditions of the surrounding TME are all significant variables able to impact the influence mutant p53 exerts over immune responses.

Efforts to target mutant p53 hold promise for intervening in its immune-regulatory function. These include the use of mutant p53 specific vaccines and autoantibodies, the reactivation of wt p53 in cellular components of the TME, and disrupting regulatory axes involving mutant p53. Overall, the complexity of mutant p53 interactions in the tumour niche convolute and limit the application of these proposed therapies, and further understanding of mutant p53 GOF is needed to fully appreciate the therapeutic capacity of targeting mutant p53 immune and inflammatory pathways.

## Figures and Tables

**Figure 1 ijms-21-03452-f001:**
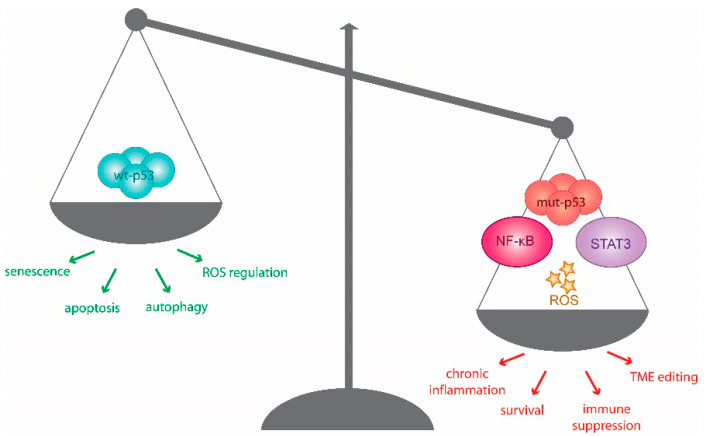
Mutant p53 feeds off and fuels inflammation. Wild-type p53 (Wt p53) contributes to effective tumour immunity through its roles in senescence, apoptosis, autophagy and ROS regulation, as well as established roles in regulating immune response. Mutant p53 is able to subvert these effects through dominant negative effects and novel tumourigenic functions mediated through the inflammatory pathways of NF-κB and signal transducer and activator of transcription 3 (STAT3). The excessive production of reactive oxygen species (ROS) can fuel inflammatory microenvironments without the regulation of wt p53, consequently feeding mutant p53 function and tipping the scales toward tumorigenesis. It is worth noting that innate immunity is not included in this model and is discussed later on.

**Figure 2 ijms-21-03452-f002:**
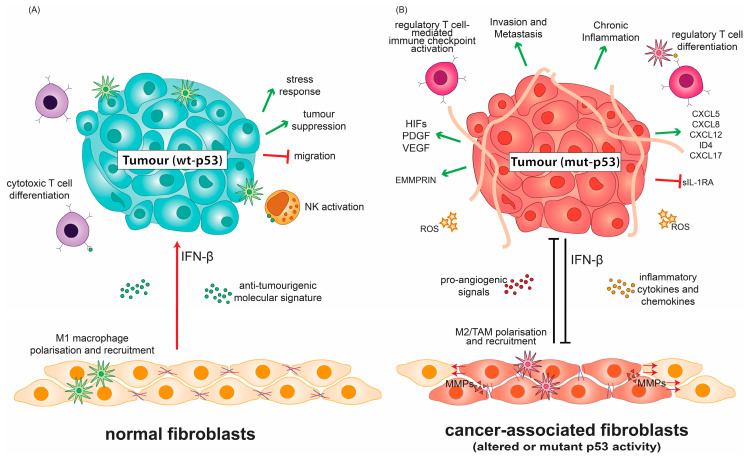
Mutant p53 alters its immunogenic niche. The tumour microenvironment is comprised of cellular and molecular components which shape the immune niche of the growing tumour. (**A**) Normal fibroblasts help mediate immunosurveillance by modulating anti-tumour immune cell infiltration and employing the IFN-β/p53 axis of regulation to activate anti-tumour responses. (**B**) Mutant p53 in the tumour alters the crosstalk between the tumour and its microenvironment, suppressing IFN-β signalling and supporting pro-angiogenic signalling. This altered crosstalk converts normal fibroblasts into tumour-supporting cancer-associated fibroblasts with altered p53 functionality. The altered transcriptional program of cancer-associated fibroblasts (CAFs) supports pro-tumorigenic signalling and extracellular matrix remodelling. The demographic and functionality of infiltrating immune cells is consequently shifted to favour immunosuppression, ultimately leading to a chronically inflamed environment and enhanced migration and invasion.

**Figure 3 ijms-21-03452-f003:**
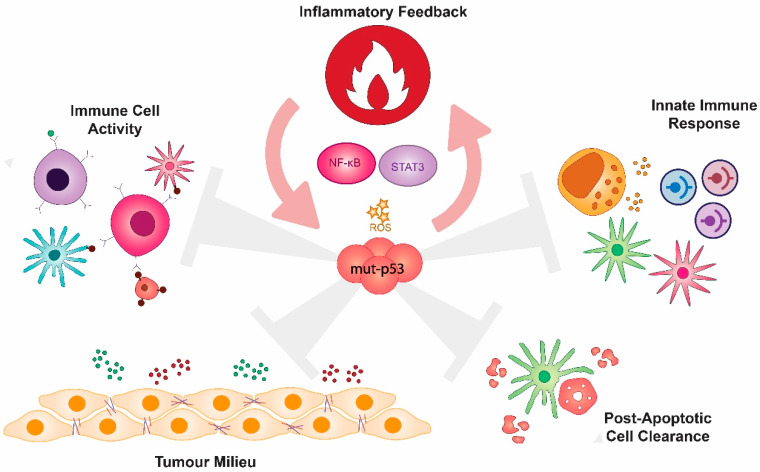
Mutant p53 impacts on the immune and inflammatory hallmarks of cancer. Mutant p53 responds to and contributes to cancer-associated chronic inflammation which facilitates several immune gain-of-functions. These are highlighted by the mutant p53-mediated alteration of the tumour milieu (pro-invasive extracellular matrix structure, cancer-associated fibroblast activity, tumour-tolerant immune cell infiltrate and chemical signatures), disabling of the innate immune response through aberrant toll-like receptor signalling, the inhibition of cell-mediated cancer immunity and potentially disrupting post-apoptotic cell clearance.

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
