# Peer review of "P53: A Guardian of Immunity Becomes Its Saboteur through Mutation"

_ijms, 2020, doi:10.3390/ijms21103452_

Round 1

Reviewer 1 Report

This is a highly focused review on the topic of mutant p53s (gain of function mutants) and their interactions with the immune system in various settings. While the topic may appear rather narrow, the research field is large. This is therefore a rather extensive review with many details and references. I find it very interesting to read and the style and grammar are excellent. Although the paper is a bit packed with information and therefore hard to digest at once. It will clearly be of interest for researchers in the p53 field. I would perhaps liked to some more discussion and perhaps some points of speculation here and there.

I find the manuscript almost ready for publication but I would like to ask the authors 1) to change the numbering of the sub sections that has gone wrong in formatting. That is, it is always section 1 or subsection 1.1.1. That was very confusing.

2) The second point is that I feel the SASP phenotype and senescence are not so much discussed. Are there any studies on mutant p53 and SASP, senescence and ageing? If so, please include that, if not that is fine as it is.

Author Response

We thank both reviewers for their positive comments on our review, and for their constructive concerns which we addressed below point by point, and as tracked changes in the text.

1) The sectioning has been revised according to the original submission. As this occurred during the reformatting at submission, we would like to ask the publisher to pay extra attention during final editing to ensure the preservation of the sectioning.

2) We mentioned senescence briefly as a non-cell-autonomous pathway regulated by p53, and added paragraphs giving a brief overview of its contribution to cancer inflammation. Because SASP generally co-opts several of the pathways already discussed in detail in this review, we kept the discussion on senescence focused on its impact on cancer, without  discussing senescence in the context of aspects of aging that are not directly related to cancer. The role of p53 and its isoforms has already been reviewed elsewhere, and new details have been added. The potential roles of mutant p53 in SASP are detailed individually when discussing the specific pathways involving NF-κB and altered signalling.

Reviewer 2 Report

This manuscript summarized the function of both wild type p53 and mutant p53 in inflammation, immunity (both innate and adaptive) and their importance in tumorigenesis, metastasis and invasion.  It is a timely and comprehensive review that discusses a complex role of p53 in modulating immunity.  Overall it is well-written.

Specific comments:

  1. The label of subsection 1; 1.1; 1.1.1 is confusing.  Please number them properly.
  2. Since both wild type p53 and mutant p53 were discussed in the review, it is important to ensure it is clearly stated the p53 is refer to wild type or mutant.  For example, in page 13, the section "p53 in post-apoptotic cell clearance", here, it is mainly talking about wild type p53.  It is suggested to use wild type p53 in the subsection title.
  3. Figure 2 was discussed in multiple subsections. It is suggested to refer it in the proper places in the text.

Author Response

We thank both reviewers for their positive comments on our review, and for their constructive concerns which we addressed below point by point, and as tracked changes in the text.

1) The sectioning has been revised according to the original submission. As this occurred during the reformatting at submission, we would like to ask the publisher to pay extra attention during final editing to ensure the preservation of the sectioning.

2) The mentioned subsection header has been revised and the document reviewed to emphasize the functional distinctions between wild-type and mutant p53

3) Discussions pertaining to Figure 2 have now been properly referred to in the text (lines 229, 234, 237, 257, 277, 291, 310, 313)